# REST Is Not Resting: REST/NRSF in Health and Disease

**DOI:** 10.3390/biom13101477

**Published:** 2023-10-02

**Authors:** Lili Jin, Ying Liu, Yifan Wu, Yi Huang, Dianbao Zhang

**Affiliations:** 1School of Life Sciences, Liaoning University, Shenyang 110036, China; lilijin@lnu.edu.cn; 2Department of Stem Cells and Regenerative Medicine, Key Laboratory of Cell Biology, National Health Commission of China, and Key Laboratory of Medical Cell Biology, Ministry of Education of China, China Medical University, Shenyang 110122, China; 2021120051@cmu.edu.cn (Y.L.); 2022120047@cmu.edu.cn (Y.W.); huangyi18004010579@163.com (Y.H.)

**Keywords:** chromatin modification, REST/NRSF, transcriptional regulation, development, cancer

## Abstract

Chromatin modifications play a crucial role in the regulation of gene expression. The repressor element-1 (RE1) silencing transcription factor (REST), also known as neuron-restrictive silencer factor (NRSF) and X2 box repressor (XBR), was found to regulate gene transcription by binding to chromatin and recruiting chromatin-modifying enzymes. Earlier studies revealed that REST plays an important role in the development and disease of the nervous system, mainly by repressing the transcription of neuron-specific genes. Subsequently, REST was found to be critical in other tissues, such as the heart, pancreas, skin, eye, and vascular. Dysregulation of REST was also found in nervous and non-nervous system cancers. In parallel, multiple strategies to target REST have been developed. In this paper, we provide a comprehensive summary of the research progress made over the past 28 years since the discovery of REST, encompassing both physiological and pathological aspects. These insights into the effects and mechanisms of REST contribute to an in-depth understanding of the transcriptional regulatory mechanisms of genes and their roles in the development and progression of disease, with a view to discovering potential therapeutic targets and intervention strategies for various related diseases.

## 1. Introduction

The repressor element-1 (RE1) silencing transcription factor (REST), also known as neuron-restrictive silencer factor (NRSF) and X2 box repressor (XBR), is a transcriptional repressor first reported in 1995. REST binds to the RE1 motif, also named neural restrictive silencing element (NRSE), and recruits corepressor complexes [1,2]. Initially identified as a regulator of neuron-specific genes in nonneuronal tissues, REST has been found to play crucial roles in neuronal development, function, and related diseases. Over nearly three decades of research, REST has been revealed to have diverse functions in other tissues such as the heart, pancreas, skin, eye, and vascular, as well as in various cancers. In this comprehensive review, we systematically summarize the research progress of REST, with a view to providing valuable insight into exploring the function of REST and the potential treatment of related diseases.

### 1.1. Gene and Protein Structure

The full-length human REST contains a DNA binding domain (DBD), an N-terminal repressor domain (NRD), and a C-terminal repressor domain (CRD) (Figure 1). REST is highly conserved in mammals, and there are various mRNA alternative splice variants predictive of multiple protein isoforms. Studies showed that REST might produce at least 45 mRNA variants encoding REST4, REST1, REST^C^, and small peptides [3]. Interestingly, Chen and Miller noted that the variable splicing of REST produces more complex protein isoforms than expected. The detections using different primers, probes, or antibodies may target different variants, and knockout (KO) or knockdown of REST can be effective against one or a few, but not all, of the specific variants. Variable splicing of REST was considered to be underestimated in some previous studies, which may have led to inconsistent data in different studies and data misinterpretation [3].

### 1.2. Nuclear Location

The nuclear location of transcription factors is essential for its modulatory function. Classically, REST binds to DNA and recruits coregulators to modulate gene expression upon entering the nucleus. The deletion of residues 512–522 in the full-length REST resulted in cytosol location. However, the isoform REST4, which lacks this putative nuclear localization signal (NLS), was localized to the nucleus, while another isoform REST1, lacking zinc finger 5, was localized in the cytosol [4]. Via deletion and mutation, the NLS around zinc finger 5 was verified to be the sole NLS within REST [5]. In addition, the nuclear envelope protein REST/NRSF-interacting Lin-11, Isl-1, and Mec-3 (LIM) domain protein (RILP) served as an essential nuclear receptor for REST [6].

### 1.3. DNA Binding

The DBD containing eight noncanonical Cys2-His2 zinc fingers (zinc finger 1–8 from N-terminus to C-terminus in Figure 1B) is crucial for DNA binding [2]. The deletion of zinc finger 3 or 5 abolished the DNA binding activity of REST, and the segment of amino acids 196–207 is essential for DNA binding [7]. Generally, each Cys2-His2 zinc finger domain binds three nucleotides within DNA, and a point mutation that changes a Cys residue to an Arg residue disrupts the Cys2-His2 zinc finger. The single mutation in zinc finger 7 and combinations of mutations in any two of zinc finger 6–8 decreases DNA binding, while a single mutation in zinc finger 6 or 8 has little effect [4]. Further, the residue Cys397 plays an important role in the global folding of multiple zinc fingers domain [8].

Initially, a homologous DNA element named NRSE within SCG10, SCN2A2, and SYN1 genes was discovered to repress their transcription via REST binding in nonneuronal cells [9,10,11]. Using biochemical, bioinformatic, and high-throughput sequencing approaches, the canonical and noncanonical potential NRSE motifs within more than 3000 genes have been characterized in vertebrates, including humans, mice, rats, and fugu [12,13,14,15,16,17,18], after the cloning of complementary DNA for REST [1,2]. These investigations have provided valuable insights into the wide-ranging presence of NRSE motifs across various species. The regulatory effects of REST on the expression of target genes are dependent on specific physiological and pathological contexts, and hundreds of target genes have been validated and studied.

Concurrently, several bioinformatics approaches were developed to facilitate the analysis of REST binding. For example, Cistematic [19] and CodingMotif [20], which were employed for predicting transcription factor binding sites. In addition, there are tools such as Site Identification from Short Sequence Reads (SISSRs) [21], CisGenome [22], quantitative enrichment of sequence tags (QuEST) [23], USeq [24], Motif Identification for ChIP-Seq Analysis (MICSA) [25], Genomic Regions Enrichment of Annotations Tool (GREAT) [26], and NEXT-peak [27] that have been developed specifically for the analysis of high-throughput chromatin immunoprecipitation (ChIP) data. These computational tools assist researchers in identifying and characterizing REST binding sites in a genome-wide context, providing valuable insights into the regulatory mechanisms and functional implications of REST.

### 1.4. Coregulator Recruitment

The transcriptional regulation mediated by REST requires the assembly of multimeric complexes [28]. Sin3 is constitutively required for REST repression, while CoREST is recruited for more specialized repressor functions [29] (Figure 2). The NRD of REST recruits Sin3A and Sin3B, which interact with HDAC1/2, HDAC4/5, and other adaptors to form a complex [30,31,32,33,34]. The HDACs remove acetyl groups from histone to silent chromatin. Nevertheless, the functions of the other adaptor proteins are largely not clear [35]. The short hydrophobic a-helix structure of NRD is captured in the cleft of the PAH1 domain of Sin3B, and PAH1 is highly conserved among Sin3A and Sin3B [36,37]. Further, the Sin3 complex is also involved in transcriptional activation [32]. Sin3A recruits Tet1 to the PAH1 domain, and Sin3 interaction domains in Tet1/Tet3 are highly conserved [38]. In the mouse retina, REST was found to mediate transcriptional activation via recruiting Tet3 to DNA for 5-hydroxymethylcytosine generation followed by NSD3-mediated H3K36 trimethylation, but no classical repressor complex protein was identified [39]. Thus, the dependency of REST-mediated gene activation on the Sin3-dependent remains enigmatic. In addition, REST recruits small CTD phosphatases (SCPs) to neuronal genes to reduce RNA polymerase II activity, and the responsible domain for SCP is still unclear [40].

The CRD containing a zinc finger (zinc finger 9) recruits CoREST as a beacon for the recruitment of repressors [41,42], including HDAC1/2, LSD1, BRG1, CtBP, G9a, SUV39H1, HP1 and MeCP2 [43]. HDACs remove acetyl groups on H3 and H4, and H3K9 deacetylation triggers localized domains of histone methyltransferase G9a-mediated H3K9 methylation [44]. The K9 residues methylated by G9a and another histone, H3K9 methyltransferase SUV39H1, are binding sites for heterochromatin protein 1 (HP1), which causes chromatin condensation [42]. Further, REST (amino acids 141–600) recruits G9a and interacts with Mediator via MED19/MED26 and MED12 subunits, respectively [45,46]. The histone demethylase LSD1 is involved in reducing H3K4 methylation [47]. The chromatin-remodeling enzyme BRG1 increases H4K8 acetylation, which stabilizes the interaction between REST and chromatin [48]. CtBP is a corepressor of REST, and the reduction of glycolysis-derived NADH promotes the recruitment of CtBP, leading to histone deacetylation [49,50]. In certain genes during neuron differentiation, methyl-CpG-binding protein MeCP2 and the CoREST repressor complex bind to methylated DNA, resulting in gene repression independently of REST [51].

## 2. REST in Nervous System

### 2.1. Embryogenesis and Neurogenesis

During mouse embryogenesis, REST was found to be required in repressing neuronal gene expression in both non-neural and neural precursor tissues [52]. The postnatal development of the hippocampus, especially the formation of the subgranular zone (SGZ), was influenced by conditional knockout (cKO) of REST in mice [53]. During early development, the loss of REST induces global hypermethylation [54,55], and the deregulation of REST was found to be an early event in Down syndrome (DS) [56,57]. In neural progenitor cells, REST was under the regulation of Wnt signaling, which controls various aspects of early development [58,59]. In zebrafish, REST serves as a regulator of hedgehog signaling, and it is required for gastrulation and neurogenesis during embryogenesis [60,61]. Additionally, REST is required for the ectodermal cell fates in *Xenopus* [62]. In addition, REST was involved in the inhibition of neurogenesis in dorsal root ganglion (DRG) by NO [63]. However, zebrafish with REST mutation generated by zinc-finger nuclease targeting underwent largely normal neurogenesis [64], and they have behavioral abnormalities such as erratic swimming and abnormal spatial preferences in adults and hypoactivity in larva [65]. In REST cKO mice, REST plays a role in the suppression of neuron genes in cultured neuronal and non-neuronal cells. However, target gene expression was not altered upon REST KO in vivo and the mice are apparently normal and grow into adults [66]. Thus, REST plays important roles in embryo development and neurogenesis [67], and its absence could be partially compensated, with the mechanisms involved being unclear.

### 2.2. Neuronal Differentiation

REST is a critical regulator of differentiation in embryonic stem cells (ESCs), neural stem cells (NSCs), and mature cell types [18,51]. In ESCs, a single REST allele disruption reduced alkaline phosphatase activity and several pluripotency-associated genes expression [68]. However, partial or complete loss of REST could not abrogate differentiation potential as reflected by marker gene expression [69]. REST knockdown in mouse ESCs could induce neuronal lineage differentiation [69], and its knockdown in human ESCs increased cell survival and the expression of mesendoderm differentiation markers [70]. Further, REST-mediated antenatal-arsenic-induced NSC function dysregulation led to deficits in differentiation in adults [71].

The repression of neuron-specific genes by REST in nonneuronal tissues highlights its crucial function during neuronal differentiation. Using ChIP-seq data mining, REST was found to mediate more than 2000 gene silencing in ESCs but not in ESC-derived neurons [72]. REST is critical for maintaining the NSCs pool, and the activation of REST target genes by REST-VP16 transgene was sufficient to induce neuronal differentiation in NSCs [73,74]. The nuclear REST was reduced during neuronal differentiation of embryonic cells, and its target genes N-methyl-D-aspartate receptor subunit 1 (NR1) and tyrosine hydroxylase (TH) were upregulated [75,76]. Further, REST plays stage-specific roles during NSC-mediated astrocyte (AS) and oligodendrocyte (OL) lineage specification, OL lineage maturation, and myelination [77,78].

The master regulatory effects of REST offer a promising strategy for the induction of neuronal differentiation. REST silencing induces neuronal differentiation of human mesenchymal stem cells (MSCs) and rat hippocampal progenitors (AHPs), and secretogranin II (Scg2) was a critical secreted REST target responsible for non-cell-autonomous phenotype [79,80]. Recently, the neuronal differentiation of human MSCs was achieved using Crispr/Cas9 targeting REST [81].

### 2.3. Neuronal Survival

REST expression is typically low in differentiated neurons, whereas its elevation frequently exhibits protective effects. For instance, the mice with neural crest-specific KO of REST presented aberrations of the myenteric plexus, leading to neonatal lethality [82]. In differentiated neural cells, overexpression of REST promoted proliferation and disrupted neurosecretion via the REST-TSC2-β-catenin signaling pathway [83]. BMP/RA-inducible neural-specific protein 1 (BRINP1) was a negative regulator of cell cycle progression, and its promotor activity was inhibited by the RE1 motif within its promotor [84]. REST inhibited BAF53a to quit mitosis by targeting miR-9* and miR-124 in post-mitotic neurons [85]. In addition, BNDF transcription was regulated by REST and its competitive isoform REST4 to regulate various neuronal cell behaviors, including cell survival [86]. In addition, ethanol was found to induce NR2B expression in fetal cortical neurons by regulating REST [87]. The augment of REST binding activity contributes to the depression of differentiation and neurogenesis induced by ethanol exposure [88,89]. REST protects the developing brain from the detrimental effects of ethanol, indicating it as a potential target for fetal alcohol syndrome treatment [90].

The expression of REST in SH-SY5Y cells was induced by polychlorinated biphenyls (PCBs), and ERK2/Sp1/Sp3/REST signaling mediated the neuronal toxicity of PCBs [91]. REST was increased during monosodium-glutamate-induced excitotoxicity in rats [92]. REST could protect cath.-a-differentiated (CAD) neuronal cells from Mn-induced toxicity by enhancing the expression of the dopamine-synthesizing enzyme tyrosine hydroxylase (TH) [93]. The dual roles of REST in the mediation of neuroprotection and neurotoxicity suggest REST as a critical regulator in neural survival.

### 2.4. Neuronal Transmission and Synaptic Plasticity

REST maintains neuronal function balance by dynamically regulating the synaptic efficiency. To facilitate neurite outgrowth, REST restricted the cell adhesion molecule L1 in the embryonic nervous system [94] and positively regulated the exocytosis of enlargeosomes [95]. However, the constitutive expression of REST led to axon guidance mistakes in the spinal cord of chicken embryos, with the depression of the target genes N-tubulin and Ng-CAM [96]. Syn1 was negatively regulated by REST to modulate synaptogenesis and neurotransmitter release [97], and Sec6 has a REST binding site and function on synapse formation and synaptic plasticity [98]. Further, the synaptic function is altered in an inflammatory environment involved in various neurodegenerative pathologies [99]. The activation of REST was found to mediate the synaptic scaling in the presence of neuroinflammation induced by IL-1β [100]. REST repressed target genes, including NPAS4 and BNDF, to regulate inhibitory synapse gene expression, thereby maintaining the excitatory/inhibitory balance in neurons [101,102]. Additionally, in NMB cells, REST inhibited VGF indicating its involvement in hippocampal synaptic activity [103].

REST protects neurons from hyperexcitability via modulating the transcriptional activity of voltage-gated ion channel genes. REST elevated by hyperactivity inhibited Nav1.2 expression to maintain homeostasis, and silencing of REST disrupted this homeostatic response [104]. Another major sodium channel, SCN8A (Nav1.6), was also found to be under the control of REST [105]. Various Ca^2+^ signaling was repressed by REST differentially [106]. The functional expression of P/Q-type channels was increased by REST silencing in GN11 neuron cells [107], while the promotor activity of N-type Ca^2+^ channel alpha 1B was not influenced by REST in NS20Y cells [108]. Recently, REST was found to control spatial K buffering in primary cortical astrocytes, impairing neuronal activity [109].

Consistently, REST presented distinct repressive actions on the transcriptional activity of neurotransmitter and receptor genes. The expression of choline acetyltransferase (ChAT) was restricted by REST to neuronal cells [110]. REST controlled tryptophan hydroxylase 2 (TPH2) expression to modulate the synthesis of neuronal serotonin [111]. The discovered receptors under the control of REST include neural-specific m4 muscarinic acetylcholine receptor (mAChR) [112,113], gamma2 subunits of the type A receptors for gamma-aminobutyric acid (GABA) [114], N-methyl-D-aspartate (NMDA) receptor subunit type I (NR1) [115], 5-HT1A receptor [116], AMPA glutamate receptor subunit GluR2 [117], and adrenergic α2C receptor (ADRA2C) [118].

### 2.5. Pain

REST regulates μ-opioid receptor (MOR) transcriptional activity [119,120,121], contributing to MOR agonist remifentanil inducing postoperative hyperalgesia [122]. PACAP was regulated by REST to mediate neuropathic pain after spinal nerve injury [123]. The negative regulation of NR2B by REST was involved in bone cancer pain [124]. The repression of MOR by REST reduced morphine analgesia in sarcoma-induced bone cancer pain [125]. REST mediated diabetic neuropathic pain in db/db mice [126]. REST also plays a major role in the acute-to-chronic pain transition upon nerve injury [127]. REST KO reverted injury-induced hyperalgesia, and REST regulates peripheral somatosensory neuron programming, leading to chronic pain [128]. REST silenced K(v)4.3 in DRG nerve injury [129]. Nerve injury elevated REST, which silenced MOP and Na(v)1.8 epigenetically [130]. REST enhances some pain, and targeting REST to release its inhibition of target genes provides a novel perspective for analgesia.

### 2.6. Neuroendocrine

The cocaine- and amphetamine-regulated transcript (CART) peptide is under strict control of REST in neuroendocrine cells [131]. The neurosecretory process involving dense-core vesicles (DCVs) was governed by REST in PC12 and astrocyte cells [132,133]. The depression of REST by Kaposi’s sarcoma-associated herpesvirus (KSHV) latent open reading frame K12 (ORFK12) gene (kaposin A) regulated neuroendocrine gene expression in infected endothelial cells [134]. Further, the proprotein convertase PC2 was regulated in nonneuroendocrine cells [135]. Interestingly, REST could repress the transcription of corticotropin releasing hormone (CRH) and also enhance the transcription via an NRSE-independent mechanism [136].

### 2.7. Intelligence and Memory

Compared with mice, human-specific genes with REST occupancy were enriched in learning and memory functions [137,138]. In early life adversity (ELA) rats, REST contributes to the memory deficits and its blockage rescued spatial memory [139]. Additionally, in APP/PS1 mice, REST is essential for the maintenance of memory performance during aging [140]. The elevated REST by Pb explore inhibited SV2C affected synaptic plasticity, leading to impairment of learning and memory [141]. During the memory impairment induced by a flame retardant PBDE-209, REST was decreased and the inhibition of NR1 was released. CDRI-08, the ethanolic extract of a nootropic plant *Bacopa monnieri*, could attenuate the impairment and restore the binding of REST to NR1 promotor [142]. Interestingly, the early acute exposure to PBDE-209 interferes with the regulation of NMDAR1 by (cAMP-response element-binding protein) CREB and REST, however, long-term effects persist only in young males that could be associated with cognitive impairment [143]. Further, the reasoning skill were not affected by the VNTR and rs2228991 variants of REST in a young Chinese Han population [144].

### 2.8. Aging and Alzheimer’s Disease

The REST-RE1 machinery is well maintained during aging in rats [145]. By regulating autophagy and senescence in neurons, REST plays a protective role during physiological aging [146]. The homologous gene of REST in *C. elegans*, Spr3, is crucial for lifespan regulation, as its mutation and silencing led to shortened lifespan [147]. REST was elevated during neuronal senescence induced by high glucose and palmitic acid [148]. Upon peripheral nerve injury, REST was elevated in young mice but not in aged mice [149]. In Alzheimer’s disease (AD), the abnormally high REST affected ChAT expression, which is a major biochemical disorder in AD [150,151]. In human cortical and hippocampal neurons, REST was elevated during aging, but it was lost in mild cognitive impairment and AD. REST protects neurons from oxidative stress and amyloid β-protein toxicity [152]. Moreover, REST levels were also found to be associated with cognitive preservation and longevity [152].

### 2.9. Parkinson’s Disease

Similar to AD, REST enters the nucleus of aged dopaminergic neurons, and it is mostly absent in neurons of Parkinson’s disease (PD) [153]. REST and REST4 were increased in the neurotoxin 1-methyl-4-phenyl-pyridinium ion (MPP^+^) induced PD cell model [154]. REST KO mice presented higher sensitivity to the dopaminergic neurotoxin MPTP [155], while the neuroprotective effects of Trichostatin A (TSA) were mediated by REST in MPTP PD models [156]. REST deficiency aggravated dopaminergic neurodegeneration and impaired neurogenesis in mice with MPTP-induced PD [157,158]. Taken together, these findings suggest that REST was a negative modulator of neurogenesis and a pro-survival factor in the context of PD.

### 2.10. Huntington’s Disease

In Huntington’s disease (HD), the increased binding of REST to RE1 reduced its target gene expression. The binding was regulated by huntingtin, and the control was lost in HD, leading to the loss of neuronal gene expression [159,160,161]. The blockage of REST binding rescued BDNF [162]. Huntingtin and REST formed a complex to facilitate REST translocation to the nucleus [163]. The transcriptional regulation of REST by huntingtin interacting protein 1 protein interactor (HIPPI) contributed to the deregulation of transcription in HD [164]. The exon-3 skipping triggered by antisense oligonucleotides (ASOs) treatment rescued neuronal genes [165]. Additionally, the inhibition of mGluR5 could correct REST signaling via Wnt pathway [166]. Thus, targeting REST is a potential treatment for HD.

### 2.11. Epilepsy

The role of REST in epilepsy is controversial, and in the literature, opposite views coexist. The elevated REST during aging was abolished in patients with drug-resistant mesial temporal lobe epilepsy (MTLE) [167]. REST cKO exhibited dramatically accelerated seizure progression and prolonged afterdischarge duration [168]. However, several investigations suggested that REST is proepileptogenic. REST cKO mice showed higher resistance to convulsions induced by PTZ [169]. The interference with the chromatin binding of REST rescued spatial memory impaired by febrile status epilepticus [170]. In kainic acid-induced status epilepticus, the repression of the HCN1 channel was mediated by REST binding [171], and mild inactivation of REST reduces susceptibility [172]. The antiepileptic effect of 2-deoxy-d-glucose (2DG), which inhibits BDNF and its receptor TrkB to reduce the progression of kindling, was mediated by REST [50], and this effect was abolished in REST cKO mice [173]. The mutation of RILP could block its interaction with REST, contributing to the progressive myoclonus epilepsy progression [174]. REST and REST4 controlled the expression of preprotachykinin-A (TAC1), which encodes the neuropeptide substance P [175,176,177]. REST mediated the contribution of Sit1 to epileptogenesis [178]. Sirt1 reduced miR-124, which targeted REST mRNA [179]. These conflicting data suggest that REST plays a role in the occurrence and development of epilepsy, and further research is needed to understand the mechanism of regulatory differences in different models.

### 2.12. Ischemia

In ischemic damage to neurons, REST appears more like a mediator. Global ischemia induced REST expression to repress GluR2 and MOR-1, and REST knockdown protected neurons from ischemia-induced death [180,181,182,183]. Moreover, the elevated REST antagonized the CREB signaling on CART activation, leading to increased cell death [184]. Pyrvinium pamoate, an activator of casein kinase 1, can suppress REST expression and rescue neurons that would otherwise undergo cell death [185]. In addition, the neuron-specific miR-132 was a REST target mediating neuronal death upon ischemia [186]. It was discovered that the protective effects of melatonin on cerebral ischemia-reperfusion injury were mediated by miR-26a-5p-NRSF and JAK2-STAT3 pathway [187].

### 2.13. Psychiatric Disorders

REST contributes to genome-wide epigenetic changes and is implicated in schizophrenia [188]. The aberrant increase of REST at the Grin2b promoter caused deficient synaptic physiology and PFC-dependent cognitive dysfunction, a hallmark of schizophrenia [189]. THP2 is the rate-limiting enzyme for 5-HT, and it is under the regulation of REST [190]. The mood-stabilizing drugs lithium and sodium valproate have been shown to regulate REST expression [191,192]. miR-9 targeted REST and PDYN, and miR-9 was targeted by REST [193]. REST was elevated in care-augmented rats, and its target CRH was inhibited [194]. The compound C737 mimics the helical structure of REST to regulate target genes involved in neuronal function and pain modulation and reduce stress-induced weight loss in female Tupaia [195]. REST signaling contributes to stress resilience [196]. The improvement in diabetes mellitus-related cognitive impairment by 9-PAHSA was mediated by REST [197]. Repression of the aberrantly increased activity of REST might be beneficial in the prevention and treatment of psychiatric disorders.

## 3. REST in Other Systems

### 3.1. Heart

REST regulates heart development, structure, and function [198]. During embryonic heart development, the elevated REST inhibited an embryonic cardiac gene HCN4 [199,200,201,202]. The training-induced REST contributes to the pathological heart rate adaptation to exercise training by reducing HCN4 [203]. REST chronically represses T channels in cardiomyocytes [204]. REST also maintains low alpha 1H expression [205] and regulates CACNA1H encoding alpha-subunit of Cav3.2 [206], thereby regulating aldosterone and cortisol production, which are independent predictors of mortality risk for heart failure patients. Gαo, an inhibitory G protein encoded by GNAO1, was regulated by REST. In several mouse models of heart failure, Gαo is decreased in the ventricles, leading to increased surface sarcolemmal L-type Ca^2+^ channel activity and impairing Ca^2+^ handling in ventricular myocytes, resulting in cardiac dysfunction [207]. REST contributes to atrial fibrillation-associated remodeling of APD and K^+^ channel expression in cardiomyocytes [208]. In cardiac hypertrophy, depressed REST releases ANP and BNP [33,209]. ANP was under the regulation of REST in ventricular myocytes and is involved in the endothelin 1-induced increase of ANP [210]. Additionally, the dominant-negative REST induces hypertrophy by targeting TRP1 and regulating calcium entry [211]. REST is negatively regulated by Zfp90 development of cardiac dysfunction [212]. The important role of REST in cardiac physiology and pathology suggests it as a potential target for the treatment of heart diseases.

### 3.2. Pancreas

REST is absent in insulinoma, insulin- and glucagon-producing cells [213]. Similar to neuronal differentiation, REST is expressed in pancreas progenitors and decreased in differentiated endocrine cells [214]. RE1 motif was found within various genes involved in pancreas development, including pdx-1, Beta2/NeuroD, and pax4 [215]. Cx36 is also controlled by REST in insulin-producing cells [216]. In addition, the rs2518719 SNP could alter the RE1 motif involving pancreatic neuroendocrine tumor progression [217]. It was revealed that the REST silencing could induce human amniotic fluid-derived stem cells (hAFSCs) and bone marrow-derived mesenchymal stem cells (bmMSCs) to differentiate into insulin-producing cells [218,219]. The similar pattern of REST regulation in the pancreas to that of neurons inspired us to focus on their similarities and differences to help build upon the well-studied foundations of REST in the nervous system and provide insights into its role in pancreatic development and function.

### 3.3. Skin

REST is expressed in skin keratinocytes, and NRP1 was found to be under its regulation to promote cell migration by blocking the inhibition of semaphorin 3A [220]. During the early neural crest stage, REST is essential for melanoblast development [221]. In melanoma, the decreased REST/RE1 activity leads to aberrant metabotropic glutamate receptor 1 (mGluR1) expression, whose ectopic expression in mouse melanocytes was sufficient to induce melanoma development [222]. These findings provide valuable insights into the functions of REST in the skin, although much more research is needed to fully understand its role in skin biology and pathology.

### 3.4. Eye

The eye, being a specialized structure derived from the skin, also relies on the proper regulation of REST. In REST cKO mice generated by the Sox1-Cre allele and the floxed REST gene, REST was excised in the early neural tissues, including the lens and retinal primordia. The slightly reduced proliferation of lens epithelial cells after birth was observed, and vacuoles formed without inducing cell apoptosis. The augmented Notch signaling and the reduced lens fiber regulator genes may contribute [223]. The REST-like gene in Drosophila was named Chn, and it promotes D1 for photoreceptor cell development [224]. This further emphasizes the importance of REST as a regulator in the development and function of eye structures. Collectively, these findings indicate the role of REST in lens formation and the maintenance of the overall structure and function of the eye.

### 3.5. Vascular

In vascular tissue, REST was found to regulate the proliferation of vascular smooth muscle cells by targeting the K(Ca)3.1 (IKCa) potassium channel. REST was downregulated when there was cellular proliferation, indicating an inverse relationship with IKCa [225]. In addition, REST declines in the medial smooth muscle of atherosclerotic carotids, and REST overexpression inhibits smooth muscle migration. Inhibition of REST and overexpression of IKCa determine smooth muscle and matrix content of atherosclerotic lesions [226].

## 4. REST in Cancer

### 4.1. Nervous System Cancer

The dysregulation of REST was found in various nervous system cancers, often playing promoting roles. In neuroblastoma cells, REST was found to be reduced during differentiation [227]. REST was downregulated in mouse neuroblastoma, leading to increased neurite length [228]. It was found that the loss of lncRNA neuroblastoma-associated transcript-1 (NBAT-1) contributed to aggressive neuroblastoma by activating REST [229]. REST is highly expressed in glioma, and its interference inhibits tumors [230]. REST mediated the inhibition of proliferation of glioma by pioglitazone [231]. In medulloblastoma, REST promotes cancer progression via epigenetic modification and AKT activation [232]. The abnormal expression of REST in NSCs blocks their differentiation and promotes medulloblastoma formation [233]. The transient expression of a competitive form of REST, REST-VP16, inhibited medulloblastoma [234,235]. Additionally, inhibiting REST via LSD1 inhibition inhibited medulloblastoma migration [236]. Moreover, the elevated REST promotes vascular growth via autocrine and paracrine [237]. The frequent neuron-specific splicing of REST is involved in neuroblastoma progression [238], while REST activity was not affected by its mutations [239]. In certain cell lines, such as NS20Y and NIE115, REST4 has been identified as the major transcript [240]. Taken together, these findings highlight REST as a potential therapeutic target for nervous system cancers, offering opportunities for developing novel treatment approaches aimed at disrupting REST-mediated tumorigenic processes.

### 4.2. Non-Nervous System Cancer

The regulation pattern of REST varies in different types of lung cancers. In small-cell lung cancer cells (SCLCs), A splice variant of REST with a 50-bp insert predicting a truncated REST was highly expressed [241]. The decreased REST permits its target gene expression, including glycine receptor alpha1 (GLRA1) subunit, migration-promoting gene ITPKA [242,243,244], leading to de-regulation of AKT activation, promoting malignant progression [245]. The altered REST splicing was found to mediate the anti-tumor effects of SRRM4 targeting in SCLC [246]. Further, miR-9 inhibits the proliferation and migration of lung squamous cell carcinoma cells by blocking REST/EGFR signaling [247]. However, in neuroendocrine lung cancer, REST mediates neuroendocrine to non-neuroendocrine fate switch to play a pro-tumorigenic role [248]. REST-regulated SCG3 provides a sensitive prognostic biomarker for neuroendocrine lung cancer [249]. Similarly, in human non-small cell lung carcinoma cells (NSCLCs), depression of REST targets contributes to tumorigenicity [250]. REST also mediates nicotine-mediated invasion and migration [251]. REST is a potential marker for the diagnosis of lung cancer types and a potential target for the treatment of some types of lung cancers. These findings suggest that REST can serve as a potential marker for diagnosing different types of lung cancer and as a potential therapeutic target for specific lung cancer subtypes.

Unlike in nervous system cancers, the loss of REST repression occurs in most non-nervous system cancers. In colorectal cancer, REST plays a suppressive role. A frameshift mutation of REST encodes a dominantly acting truncation activating PI3K signaling [252], and rs968697 polymorphism in the HMGA2 gene influences REST binding [253]. In hepatocellular carcinoma (HCC), REST was found to be elevated in 27 of 49 human tissue samples, and it plays a key role in promoting diethylnitrosamine (DEN)-induced HCC initiation [254]. However, another study revealed that REST is decreased in cholangiocellular carcinoma but not in HCC when compared with normal liver tissues [255]. In breast cancer, REST was decreased, contributing to pathogenesis [256]. Additionally, REST is required for beta-estradiol (E2) stimulation of the cell cycle [257]. In Ewing sarcoma, EWSR1 interacted with REST, and EWSR1 reduced genes are a subset of neuronal genes that contain RE1 [258]. In uterine fibroids, REST represses GPR10 in the normal myometrium, and that the loss of REST in fibroids permits GPR10 expression [259]. In oral cancer, REST plays an important role in the cell survival by regulating the mTOR signaling pathway [260]. In the human prostate carcinoma cell line LNCaP, REST repressed IB1/JIP-1 to regulate the neuronal phenotype and JNK signaling pathway [261]. Furthermore, altered REST splicing was also found to mediate the anti-tumor effects of SRRM4 targeting in prostate cancer cells [246]. A recent study revealed REST diminution in premalignant lesions, squamous cell carcinoma (SCC), endocervical adenocarcinoma (ADC), and the cancerous cell lines SiHa and HeLa [262]. These investigations suggest that REST may be a hub that extensively regulates tumor occurrence and development and can serve as a diagnostic and therapeutic target. These investigations suggest that REST may serve as a central regulator in tumor occurrence and development, making it a potential diagnostic and therapeutic target in various types of cancer.

## 5. Targeting REST

Considering the important role of REST in health and disease, multiple strategies have been developed to inhibit or activate REST. Small interfering RNA (siRNA) is widely used to knockdown gene expression, and siRNA targeting REST could inhibit GBM cells and ovarian cancer cells [263,264,265]. In addition, the siRNA delivered by PEI@Fe_3_O_4_ nanoparticles induced MSC differentiation into insulin-producing cells [266]. In contrast to siRNA, the nuclear-localized small modulatory double-stranded RNA (smRNA) encoding NRSE sequences has been developed to activate REST target genes via smRNA/REST interaction, promoting neuronal differentiation [267,268]. Another approach involves the use of the recombinant transcription factor REST-VP16, which was constructed by replacing the repressor domains of REST with the activation domain of a viral activator VP16. It could induce neuronal differentiation in NSCs, myoblasts, and possibly other stem cells [74,269,270]. Small molecules have also been identified as REST modulators. For instance, the small molecule mS-11 was designed based on the mSin3-binding helix structure of REST. It inhibits mSin3-REST binding to reverse pain and autism in mice models [271,272,273]. The quinolone-like compound 91 (C91) was discovered by virtual screening. It reduces mSIN3b nuclear entry and occupancy at NRSE within BNDF to restore BDNF levels in HD cells [274]. Sertraline, chlorprothixene, and chlorpromazine were also found to interact with the REST-binding site of mSin3 to inhibit medulloblastoma [275]. Further, three 2-aminothiazole derivatives were found to modulate NRSE silencing activity [276]. Through high-throughput screening with luciferase-assay measuring REST activity, a benzoimidazole-5-carboxamide derivative X5050 was found to induce REST degradation, other than REST expression, splicing, or NRSE binding [277]. In addition, the small molecule compound C737 mimicking the helical structure of REST reduced stress-induced weight loss in female Tupaia [195]. The optogenetic approach mediated REST inhibition was achieved using the fusion of the inhibitory domains that impair REST-DNA binding or recruitment of the cofactor mSin3a to the light-sensitive Avena sativa light-oxygen-voltage-sensing 2-phototrophin 1 to modulate REST target genes [278]. Recently, an approach combining biochemical experiments and molecular dynamics simulations was applied to develop ad hoc designed RNA binding proteins to specifically target and dock to REST mRNA [279]. These diverse regulatory strategies provide a solid foundation for further investigations and drug development targeting REST, opening up novel possibilities for therapeutic interventions in REST-related conditions.

## 6. Concluding Remarks

REST is a key transcription regulator that primarily governs gene transcription via the modification of chromatin. REST plays vital roles in both physiological and pathological processes within the nervous system as well as other non-nervous systems. The imbalance of REST dynamic regulation often leads to dysplasia or diseases. Therefore, it is necessary to conduct in-depth research on its involvement in different diseases and the underlying mechanisms is imperative to gain novel insights into potential interventions for related diseases.

The patterns of interaction between REST and its target genes, as well as the mechanisms underlying gene transcription regulation, have been identified by current research. The dynamic regulation of REST expression and gene targeting during development and in various diseases remains largely unknown. However, advancements in technologies such as single cell sequencing, spatial transcriptome, structural biology, and artificial intelligence hold promise in acquiring more relevant information. Integrating these findings with cellular and molecular biology technologies will strongly enhance our understanding of the mechanism of transcription factors, including REST.

REST plays a pivotal role in early development, and its regulation exhibits robustness with upstream bias. While many investigations have focused on understanding the regulation of target genes by REST, there is still much to be discovered regarding the upstream regulatory mechanisms governing REST transcription, translation, and activity. Unraveling these aspects would contribute to the construction of comprehensive gene regulatory networks centered around REST.

Different antibodies target different antigenic epitopes, and different REST isoforms are post-translationally modified, making it difficult to discriminate between non-specific bands and REST isoforms on the basis of band position in Western blotting data. Simply cropping out bands that do not match the expected molecular weight is likely to miss a lot of important information. In addition, the use of currently unavailable REST antibodies in some studies might lead to non-reproducible or misleading results due to, for example, poor specificity [3]. Therefore, the specificity of the REST antibody and the multiple bands in the blotting are strongly recommended to be cautioned in cropping and analysis.

In conclusion, this review highlights the mechanisms of REST and the progress made in understanding its roles in health and disease. It aims to provide assistance in comprehending the mode of action and exploring potential interventions for the related diseases. REST is a fascinating gene that merits further investigation, and we eagerly anticipate more interdisciplinary research results and breakthroughs in the field of REST.

## Figures and Tables

**Figure 1 biomolecules-13-01477-f001:**
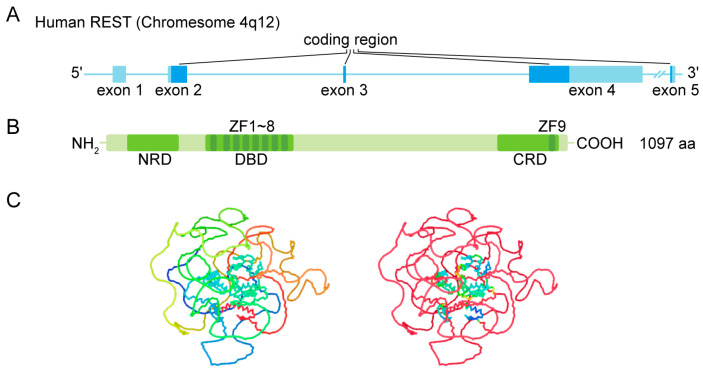
Gene structure and protein structure of REST. (**A**) Human REST is located in Chromosome 4q12, and it has five exons. The exon 5 was found to be mutually exclusive of exon 4. (**B**) The full-length human REST consists of 1097 aa, and there is a DNA binding domain (DBD), an N-terminal repressor domain (NRD), and a C-terminal repressor domain (CRD) for its regulatory activities. ZF indicates zinc finger. (**C**) The protein structure was predicated using Alphafold2.

**Figure 2 biomolecules-13-01477-f002:**
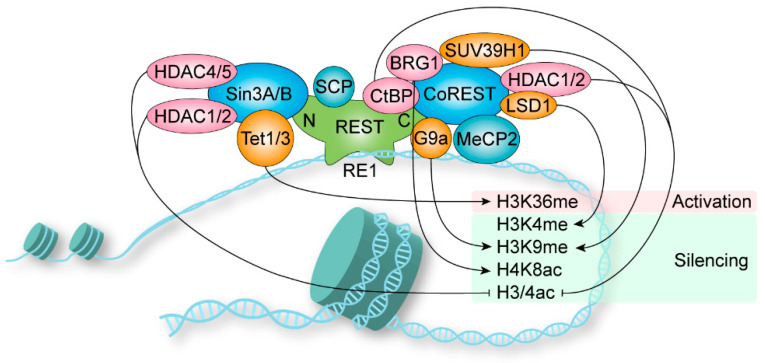
REST modulates gene activation via histone modification. REST binds to the RE1 motif within target genes and recruits Sin3A/B and CoREST as beacons for the recruitment of coregulators to form regulatory complexes. The chromatin was silenced via histone 3/4 deacetylation by HDACs, H3K9 methylation by G9a and SUV39H1, and H3K4 demethylation by LSD1 and H4K8 acetylation by BRG1. Further, the H3K36 methylation mediated by Tet1/3 might activate gene transcription.

## Data Availability

Not applicable.

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
