# Peer review of "REST Is Not Resting: REST/NRSF in Health and Disease"

_biomolecules, 2023, doi:10.3390/biom13101477_

Round 1

Reviewer 1 Report

This review article is just simply a list of previously reported articles sorted by diseases. This current simple review form by authors is not much informative. The authors just listed them in the way they want to organize. It's easy to understand if you put them in a table, but there are similar reviews by other researchers so far.

Authors has not indicated several controversial contents, by deliberately emphasizing. The review article should include the contents which are more informative if authors summarize the arguments. For example, Chen & Miller pointed out the alternative REST splicing was largely neglected by Lu et al (2014), making it necessary for a reevaluation of their findings. 

It is not correct that they mentioned “REST4 is the first reported competitive variant and the others remain to be recognized [3]”. The authors clearly have no knowledge of isoforms. Figure 1 is not correct.

In some of articles, the authors used several REST-antibodies that had been discontinued due to the uncertain specificity and the results are controversial. Authors should mention about this, making the current article much better informative. 

Some of abbreviations should be written formally and some of references should be correct. 

Author Response

Thank you for your kindly checking and professional comments. The manuscript has been improved following your helpful suggestions. The changes have been marked using Microsoft Office build-in Track Changes. The responses to the comments are as following:

Reviewer 1:

This review article is just simply a list of previously reported articles sorted by diseases. This current simple review form by authors is not much informative. The authors just listed them in the way they want to organize. It's easy to understand if you put them in a table, but there are similar reviews by other researchers so far.

Response:

Thank you for your checking of the manuscript and providing helpful comments. Our research in recent years has identified a role of REST in epithelial structure, and the manuscript is expected to be published within the year. During our research, we read most of the available papers on REST, including some reviews. These previous findings and reviews gave us an insight into the progress of research in the last 30 years since the discovery of REST and many inspirations. During the reading of these previous papers, we organized the main findings and opinions, which gradually formed this manuscript. We hope that this manuscript will be helpful to researchers who are interested in REST and its related functions.

Authors has not indicated several controversial contents, by deliberately emphasizing. The review article should include the contents which are more informative if authors summarize the arguments. For example, Chen & Miller pointed out the alternative REST splicing was largely neglected by Lu et al (2014), making it necessary for a reevaluation of their findings.

Response:

Thank you for your comments. As you pointed out, the inclusion of the controversy will make the manuscript more informative and enlightening. The revised manuscript has been modified as per your comments. The following paragraph has been added to section 1.1: Interestingly, Chen & Miller noted that the variable splicing of REST produces more complex protein isoforms than expected. The detections using different primers, probes, or antibodies may target different variants, and knockout or knockdown of REST can be effective against one or a few, but not all, of the specific variants. Variable splicing of REST was considered to be underestimated in some previous studies, which may have led to inconsistent data in different studies, and data misinterpretation (Chen, G.L.; Miller, G.M. Alternative REST Splicing Underappreciated. Eneuro 2018, 5, doi:10.1523/ENEURO.0034-18.2018.).

It is not correct that they mentioned “REST4 is the first reported competitive variant and the others remain to be recognized [3]”. The authors clearly have no knowledge of isoforms. Figure 1 is not correct.

Response:

Thank you for your comments. It has been improved to “Studies showed that REST might produce at least 45 mRNA variants encoding REST4, REST1, RESTC, and a number of small peptides, among others.” in the revised manuscript. For Figure 1, Exon 5 has been added. The description and figure legend has been modified accordingly.

In some of articles, the authors used several REST-antibodies that had been discontinued due to the uncertain specificity and the results are controversial. Authors should mention about this, making the current article much better informative.

Response:

As you pointed out, antibody specificity is indeed troubling, and there are no satisfactory available antibodies for quite a few proteins, including REST. This comment of yours has also been very helpful to us in studying REST, giving us more insight into some of the data, in terms of antibody specificity and variable splicing. Based on your suggestion, the following paragraph was added to Section 6 in order to emphasize that this issue is of concern: Different antibodies target different antigenic epitopes, and different REST isoforms are post-translationally modified, making it difficult to discriminate between non-specific bands and REST isoforms on the basis of band position in western blotting data. Simply cropping out bands that do not match the expected molecular weight is likely to miss a lot of important information. In addition, the use of currently unavailable REST antibodies in some studies might lead to non-reproducible or misleading results due to, for example, poor specificity. Therefore, the specificity of the REST antibody and the multiple bands in the blotting are strongly recommended to be cautioned in cropping and analysis.

Some of abbreviations should be written formally and some of references should be correct.

Response:

Thank you for your checking and comments. Abbreviations and references were checked and revised in the revised version of the manuscript.

Kind regards,

Dianbao Zhang

China Medical University

Reviewer 2 Report

In this review paper, Jin and colleagues provide a comprehensive overview of the functions of the transcription repressor REST/NRSF, giving details about its role under healthy and diseased conditions. After an introduction about gene and protein structure, nuclear localization and formation of transcriptional complex, the authors focus on the role of REST in the nervous system, during development, plasticity and some neuronal pathologies. After this, they describe available evidence linking REST to diseases outside of the nervous system, focusing on selected types of cancer. The last paragraph gives a short overview of the mechanisms used to target REST and modify its function in preclinical research.

REST is a protein that plays an extremely complex role both during development and in the adult organism. The authors did a very good job in summarizing and organizing available evidence in a text that is quite complete and easy to read. I appreciated their effort and I support the publication of this review. I will comment only on the aspects that more directly fall within my area of expertise.

- Chapter 2.4 ‘Neuronal transmission’: first, I would rename this section ‘Neuronal transmission and synaptic plasticity’. I would then briefly discuss the notion that REST mediates synaptic scaling in the presence of neuroinflammation (https://pubmed.ncbi.nlm.nih.gov/33589593/). This is relevant as synaptic function is altered in an inflammatory environment (https://pubmed.ncbi.nlm.nih.gov/29674955/), which is common to almost all neurodegenerative pathologies.

- The section 2.11 ‘Epilepsy’ in my opinion is not well structured and should be revised. The role of REST in epilepsy is controversial and in the literature opposite views coexist. In fact, while some papers suggest a protective role of REST against epilepsy (https://pubmed.ncbi.nlm.nih.gov/21339379/), others suggested REST is proepileptogenic (https://pubmed.ncbi.nlm.nih.gov/21905079/; https://pubmed.ncbi.nlm.nih.gov/22472570/; https://pubmed.ncbi.nlm.nih.gov/31998079/). This is explained in the text but it could be made more clear.

- In the section ‘Targeting REST’, some optogenetic-based and in silico approaches have also been published, which should be at least briefly mentioned (https://pubmed.ncbi.nlm.nih.gov/26699507/; https://pubmed.ncbi.nlm.nih.gov/32678979/).

Author Response

Thank you for your kindly checking and professional comments. The manuscript has been improved following your helpful suggestions. The changes have been marked using Microsoft Office build-in Track Changes. The responses to the comments are as following:

Reviewer 2:

In this review paper, Jin and colleagues provide a comprehensive overview of the functions of the transcription repressor REST/NRSF, giving details about its role under healthy and diseased conditions. After an introduction about gene and protein structure, nuclear localization and formation of transcriptional complex, the authors focus on the role of REST in the nervous system, during development, plasticity and some neuronal pathologies. After this, they describe available evidence linking REST to diseases outside of the nervous system, focusing on selected types of cancer. The last paragraph gives a short overview of the mechanisms used to target REST and modify its function in preclinical research.

REST is a protein that plays an extremely complex role both during development and in the adult organism. The authors did a very good job in summarizing and organizing available evidence in a text that is quite complete and easy to read. I appreciated their effort and I support the publication of this review. I will comment only on the aspects that more directly fall within my area of expertise.

- Chapter 2.4 ‘Neuronal transmission’: first, I would rename this section ‘Neuronal transmission and synaptic plasticity’. I would then briefly discuss the notion that REST mediates synaptic scaling in the presence of neuroinflammation (https://pubmed.ncbi.nlm.nih.gov/33589593/). This is relevant as synaptic function is altered in an inflammatory environment (https://pubmed.ncbi.nlm.nih.gov/29674955/), which is common to almost all neurodegenerative pathologies.

Response:

Thank you for your comments. The section title has been improved to “Neuronal transmission and synaptic plasticity” following your suggestion. The discussion of REST in synaptic scaling has been added to this section: The synaptic function is altered in an inflammatory environment involved in various neurodegenerative pathologies (https://pubmed.ncbi.nlm.nih.gov/29674955/). The activation of REST was found to mediate the synaptic scaling in the presence of neuroinflammation induced by IL-1β (https://pubmed.ncbi.nlm.nih.gov/33589593/).

- The section 2.11 ‘Epilepsy’ in my opinion is not well structured and should be revised. The role of REST in epilepsy is controversial and in the literature opposite views coexist. In fact, while some papers suggest a protective role of REST against epilepsy (https://pubmed.ncbi.nlm.nih.gov/21339379/), others suggested REST is proepileptogenic (https://pubmed.ncbi.nlm.nih.gov/21905079/; https://pubmed.ncbi.nlm.nih.gov/22472570/; https://pubmed.ncbi.nlm.nih.gov/31998079/). This is explained in the text but it could be made more clear.

Response:

Thank you for your professional and detailed comments. The section 2.11 has been revised according to your suggestions. The structure has been improved and the suggested references have been added in the revised manuscript.

- In the section ‘Targeting REST’, some optogenetic-based and in silico approaches have also been published, which should be at least briefly mentioned (https://pubmed.ncbi.nlm.nih.gov/26699507/; https://pubmed.ncbi.nlm.nih.gov/32678979/).

Response:

As you suggested, the optogenetic-based and in silico approaches have been added to the section “Targeting REST”, making the section more comprehensive: The optogenetic approach mediated REST inhibition was achieved using the fusion of the inhibitory domains that impair REST-DNA binding or recruitment of the cofactor mSin3a to the light-sensitive Avena sativa light-oxygen-voltage-sensing 2-phototrophin 1, to modulate REST target genes (https://pubmed.ncbi.nlm.nih.gov/26699507/). Recently, an approach combining biochemical experiments and molecular dynamics simulations was applied to develop ad hoc designed, RNA binding proteins to specifically target and dock to REST mRNA (https://pubmed.ncbi.nlm.nih.gov/32678979/).

Kind regards,

Dianbao Zhang

China Medical University

Round 2

Reviewer 1 Report

The authors tried to respond to the first round of reviewer’s comments. However, the authors did not mention a table of previous review comments to improve the manuscript. If a table or figure listing the molecular weight of detected REST isoforms and antibodies is not included in the manuscript, this review is just a list of several manuscripts and is difficult to understand. Some references have turned out to be contradictory.

In addition, the authors added sentences regarding the antibody and REST splicing isoforms, but did not correct the other parts accordingly. For example, “2.8 Aging and Alzheimer's disease” are contradictory within the text. This is because of ignoring the existence of isoforms and strange detection by previous antibodies, which are difficult to understand because they are not mentioned.

I appreciate the time authors spent trying to cover all the papers, but unfortunately I do not recommend publishing it as it is. This review will lead to misunderstandings regarding REST/NRSF.

Author Response

Thank you for your swift reply and processing. The response to the comments of Reviewer 1 is as following.

Comments and Suggestions for Authors:

The authors tried to respond to the first round of reviewer’s comments. However, the authors did not mention a table of previous review comments to improve the manuscript. If a table or figure listing the molecular weight of detected REST isoforms and antibodies is not included in the manuscript, this review is just a list of several manuscripts and is difficult to understand. Some references have turned out to be contradictory.

In addition, the authors added sentences regarding the antibody and REST splicing isoforms, but did not correct the other parts accordingly. For example, “2.8 Aging and Alzheimer's disease” are contradictory within the text. This is because of ignoring the existence of isoforms and strange detection by previous antibodies, which are difficult to understand because they are not mentioned.

I appreciate the time authors spent trying to cover all the papers, but unfortunately I do not recommend publishing it as it is. This review will lead to misunderstandings regarding REST/NRSF.

Response:

Thank you for your time and effort on our manuscript. Your comments are concerned with the detection of REST isoforms by various antibodies against REST, as well as some of the findings addressed in Section 2.8 Aging and Alzheimer's disease. An in-depth and enlightening discussion of this has been published (Chen, G.L.; Miller, G.M. Alternative REST Splicing Underappreciated. Eneuro 2018, 5, doi:10.1523/ENEURO.0034-18.2018). In the paper, it is discussed that different antibodies may detect different REST isoforms due to their different specificities, and that neglect of REST isoforms may lead to misinterpretation of data. These discussions raise concerns about variable splicing of REST, which is interesting. The issue of antibody specificity and variable splicing has the potential to influence the conclusions of previous studies. However, the conclusions of previous studies are not sufficiently discredited without experimental confirmation. The issues raised in the discussion are worthy of attention and research. This issue is discussed in the manuscript following your previous suggestions, and it is expected to enlighten readers without misleading. Your comments also make us more interested in the variable splicing of REST, and we intend to study the role of REST variable splicing in its regulatory function, especially in epithelial structure and function.

Kind regards,

Dianbao Zhang

China Medical University